

# Improving satellite-retrieved surface radiative fluxes in polar regions using a smart sampling approach

Van Tricht Kristof[1], Lhermitte Stef[1], Gorodetskaya Irina V.[1], and van Lipzig Nicole P.M.[1]

[1]KU Leuven - University of Leuven Department of Earth and Environmental Sciences, Celestijnenlaan 200E, Leuven 3001, Belgium

*Correspondence to:* Kristof Van Tricht (Kristof.VanTricht@kuleuven.be)

**Abstract.** The surface energy budget (SEB) of polar regions is key to understanding polar amplification of global climate change and its worldwide consequences. Yet, despite a growing network of ground-based automatic weather stations that measure the radiative components of the SEB, extensive areas remain where no ground-based observations are available. Satellite remote sensing has emerged as a potential solution to retrieve components of the SEB over remote areas, with radar and lidar aboard the CloudSat and CALIPSO satellites among the first to enable estimates of surface radiative long-wave (LW) and short-wave (SW) fluxes based on active cloud observations. However, due to the small swath footprints, combined with a return cycle of 16 days, questions raise as to how CloudSat/CALIPSO observations should be optimally sampled in order to retrieve representative fluxes for a given location. Here we present a smart sampling approach to retrieve downwelling surface radiative fluxes from CloudSat/CALIPSO observations for any given land-based point-of-interest (POI) in polar regions. The method comprises a spatial correction that allows to increase the distance between satellite footprint and POI in order to raise the satellite sampling frequency. Sampling frequency is enhanced on average from only two unique satellite overpasses each month for limited-distance sampling <10 km from the POI, to 35 satellite overpasses for the smart sampling approach. This reduces the root-mean-square errors on monthly mean flux estimates compared to ground-based measurements from 23 W m$^{-2}$ to 10 W m$^{-2}$ (LW) and from 43 W m$^{-2}$ to 14 W m$^{-2}$ (SW). The added value of the smart sampling approach is shown to be largest on finer temporal resolutions, where limited-distance sampling suffers from severely limited sampling frequencies. Finally, the methodology is illustrated for Pine Island Glacier (Antarctica) and the Greenland northern interior, remote areas where few ground-based observations are available yet where important climatic changes have been recently reported, to yield 5-day moving average timeseries of downwelling LW and SW fluxes. We conclude that the smart sampling approach may help to reduce the observational gaps that remain in polar regions to further refine the quantification of the polar SEB.

## 1 Introduction

Polar regions experience global climate change to an amplified extent compared to other areas, known as polar amplification (Holland and Bitz, 2003; IPCC, 2014), demonstrating their crucial role in earth's climate. The surface energy budget (SEB) is one of the key elements describing the climate system (Trenberth et al., 2009), and its quantification in polar regions is



therefore paramount to understand the feedback processes that cause the amplified response to global climate change (Vaughan et al., 2003; Turner, 2005; Convey et al., 2009; Kay et al., 2011; Serreze and Barry, 2011).

Different components of the local SEB can be retrieved by specialised equipment such as radiometers and spectrometers (Ohmura et al., 1998), that have led to the deployment of numerous automatic weather station (AWS) networks across both the

Arctic and the Antarctic (Steffen and Box, 2001; van den Broeke, 2004; van den Broeke et al., 2008; Ahlstrøm et al., 2008; Lazzara et al., 2012). Yet, despite the increasing amount of AWSs, the distribution of these ground-based observations of energy components remains strongly irregular with numerous blind spots over extensive areas, hindering an accurate assessment of the complete polar energy budget.

Radiative fluxes that cover the entire polar regions, including these blind spots, can potentially be retrieved from renanalysis

products such as the European Centre for Medium-Range Weather Forecasts (ECMWF) atmospheric re-analysis (ERA). Yet, the accuracy of these products in a certain area depends heavily on the amount of available observations (Dee et al., 2011), which is severely limited in large parts of the remote polar regions. This is especially true with regard to cloud observations (Kay and L'Ecuyer, 2013; Naud et al., 2014), favouring a more observation-based approach.

With the advent of satellite remote sensing, a rapidly increasing amount of data over remote regions has become available.

For the first time, an observation-based global assessment of the top-of-atmosphere (TOA) radiation budget could be conducted using satellite observations during missions such as 'Earth Radiation Budget' (ERB), 'Earth Radiation Budget Experiment' (ERBE) and later 'Clouds and the Earth's Radiant Energy System' (CERES) (Kyle et al., 1993; Barkstrom and Smith, 1986; Smith et al., 1994; Wielicki et al., 1996; Loeb et al., 2002; Gorodetskaya et al., 2006). Satellites involved in these missions carry passive radiometers that are used to retrieve broadband upwelling short-wave (SW↑) and long-wave (LW↑) radiative

fluxes at the TOA.

However, inferring the SEB from TOA observations requires thorough knowledge on atmospheric constituents and how these alter the energy exchange between earth's surface and the TOA. Clouds are one of the dominant atmospheric features that interact with radiation in polar regions (Bintanja and Van Den Broeke, 1996; Curry et al., 2000; Gorodetskaya et al., 2008; Kay et al., 2008; Bromwich et al., 2012; Van Tricht et al., 2014; Miller et al., 2015), and were for instance shown to be

responsible for a cloud radiative effect of 29.5 W m$^{-2}$ over the Greenland ice sheet (Van Tricht et al., 2016). For the retrieval of a reliable SEB by satellite remote sensing, it is therefore paramount to include proper cloud observations in the radiative transfer calculations, and radiometers from space do not provide sufficient information to that end.

After the launch of the space-based active radar and lidar instruments onboard of the CloudSat and Cloud-Aerosol Lidar and Infrared Pathfinder Satellite Observations (CALIPSO) satellites in 2006, cloud observations from space entered a new era.

The complementary nature of a cloud-penetrating radar and a lidar that does not suffer from ground reflections (Maahn et al., 2014) for observing cloud macro- and microphysical properties allows an accurate characterization of the atmospheric column (Stephens et al., 2002; Winker et al., 2009; Mace et al., 2009). In addition, active satellite remote sensing over polar regions is not compromised by weak brightness temperature differences that are inherent over snow and ice surfaces (Bromwich et al., 2012), yielding a valuable dataset for cloud studies in polar regions (Grenier et al., 2009; Kay and Gettelman, 2009; Devasthale

et al., 2011; Liu et al., 2012; Cesana et al., 2012; English et al., 2014). The Level-2 "Fluxes and Heating Rates" (2B-FLXHR-





LIDAR) product is among the first to use active remotely-sensed cloud observations to retrieve surface radiative fluxes on a global scale (Henderson et al., 2013) and has been succesfully used to study cloud impacts on the energy budget in polar regions (Kay and L'Ecuyer, 2013; Van Tricht et al., 2016; Christensen et al., 2016).

Despite the advantage of these active satellite observations, however, the swath width of CloudSat and CALIPSO, sun-synchronous polar-orbiting satellites, is small (∼1.4 km). The spatial patterns of these narrow-swath satellites therefore show numerous blind spots where no overpasses are available. At the same time, the repeat cycle of these overpasses is only once every 16 days (Winker et al., 2009). An inherent drawback of narrow-swath satellite observations therefore is a limited spatial and temporal coverage.

One way to enhance this spatial and temporal resolution is by extrapolating the narrow-swath satellite data to nearby locations, since radiative fluxes at the surface are to some degree spatially correlated (Long and Ackerman, 1995). However, this introduces a tradeoff (Fig. 1) between enhancing the spatial and temporal resolution by including more satellite overpasses from nearby regions, and decreasing the spatial representativeness of each overpass that is included. This means that increasing the maximum distance to a point for which satellite profiles are still taken into account decreases the time between subsequent overpasses, but at the same time increases the expected root-mean-square error (RMSE) between satellite retrievals further away and ground truth at the location itself (Fig. 1).

Here we present a methodology to optimize this tradeoff for estimating downwelling SW (SW↓) and LW (LW↓) radiative fluxes at any given land-based point-of-interest (POI) in the polar regions, with estimated uncertainties for each retrieval. To that end, we first investigate the regional dynamics that determine the spatial representativeness of nearby CloudSat/CALIPSO overpasses. Then, the temporal representativeness of CloudSat/CALIPSO data is quantified. This information is finally used to develop a smart sampling approach to estimate SW↓ and LW↓ radiative fluxes at any given POI without the need for external information. The methodology is evaluated based on AWS measurements at six locations and its use is illustrated for Pine Island Glacier (Antarctica) and the Greenland northern interior, that were previously blind spots where few or no information from AWSs is available, while important climatic changes have been recently reported at these locations (Jenkins et al., 2010; Nghiem et al., 2012).

## 2 Data

### 2.1 Study area and automatic weather stations

The study area comprises the land-covered regions north of 60° N (Arctic) and south of 60° S (Antarctic). For developing the methodology and evaluation purposes, retrieved radiative fluxes from CloudSat/CALIPSO are compared to ground-based fluxes measured by AWSs, including five stations from the Baseline Surface Radiation Network (BSRN) (Ohmura et al., 1998) and an AWS at the Princess Elisabeth (PE) station in Antarctica (Gorodetskaya et al., 2013, 2015) (Fig. 2). These AWSs measure broadband downwelling and upwelling SW and LW radiative fluxes at the surface using pyranometers and pyrgeometers. More information on the locations and instrument specifications of the AWSs is given in Table 1.





## 2.2 CloudSat and CALIPSO satellite observations

CloudSat and CALIPSO were launched in 2006 to globally observe clouds from a near-polar orbit. CloudSat carries the Cloud Profiling Radar instrument, a 94-GHz nadir-looking radar, while CALIPSO carries the Cloud-Aerosol Lidar with Orthogonal Polarization (CALIOP) instrument, a two-wavelength (532 nm and 1064 nm) polarization-sensitive lidar. The complementary

nature of CALIOP and CPR, with the former designed to focus on thin clouds and the latter probing thicker clouds and precipitation, allows an unprecedented three-dimensional characterization of clouds on a global scale (Stephens et al., 2009; L'Ecuyer and Jiang, 2010).

CALIOP and CPR measure the backscattered energy by cloud particles, which are then converted into vertical distributions of cloud ice and liquid water contents and effective radii, filled in by Moderate Resolution Imaging Spectroradiometer (MODIS)

radiance information (Platnick et al., 2003) when the retrieval algorithms of the active sensors fail to converge. These merged active satellite cloud observations have been succesfully used for determining the radiative importance of clouds in polar regions (e.g., Liu et al., 2012; Kay and L'Ecuyer, 2013; Van Tricht et al., 2016).

The 2B-FLXHR-LIDAR product used in this study combines satellite-retrieved cloud properties to drive the two-stream radiative transfer model BugsRad that calculates the observationally-constrained radiative broadband (SW + LW) fluxes at

126 vertical levels, including the surface (Henderson et al., 2013). Cloud observations are combined with atmospheric profiles of temperature and humidity and sea surface temperatures from ECMWF ERA-Interim reanalyses, and with surface albedo and emissivity data from the International Geosphere–Biosphere Programme (IGBP) global land surface classification. The horizontal resolution of an individual CloudSat/CALIPSO profile is about 1.4 km by 1.7 km. Subsequent profiles therefore form an overpass with a narrow swath width of 1.4 km.

## 3   Issues related to narrow-swath satellite sampling

### 3.1   Spatial representativeness

Nearby satellite overpasses are not necessarily representative for a POI. Apart from the fact that weather systems can be different when the distance between a satellite footprint and a POI becomes too large, the representativeness of narrow-swath CloudSat/CALIPSO radiative flux retrievals can also be compromised by differences in (i) surface characteristics, (ii) sun

position and TOA insolation, and (iii) altitude.

**Surface characteristics**

Radiative fluxes that are retrieved over surfaces with significantly different characteristics compared to the POI will decrease the representativeness, even for the downwelling components. SW↓ and LW↓ radiation are strongly influenced by the atmospheric state (cloud properties, temperature and humidity profiles and aerosol contents), the surface (SW albedo, LW emissivity and

temperature), and the interaction between both. In the case of SW↓ radiation, multiple reflection between the surface and clouds and hence SW↓ radiation increases substantially over highly-reflective surfaces such as snow and ice (Bintanja and





Van Den Broeke, 1996), an effect that is further aggravated by the high solar zenith angles (SZAs) in polar regions (Shupe and Intrieri, 2004). At the same time, LW↓ radiation is affected by surface temperatures and LW emissivity that directly influence the atmospheric state. For example, water bodies emit more LW radiation which warms the atmosphere in addition to higher moisture fluxes as well. The resulting warmer and moister atmosphere yields higher LW↓ radiative fluxes compared to an atmosphere over snow-covered surfaces, which is cooler and dryer (Marty et al., 2002). However, significant differences can arise even if both the POI and satellite overpasses are situated over land due to the large possible variety of surface characteristics. For example, rock-covered surfaces have a much lower albedo in contrast to snow/ice-covered surfaces with significant consequences for the SW↓ and LW↓ radiative fluxes. Surface albedo is therefore a useful parameter to descriminate between different surface types that can influence the SW↓ and LW↓ radiative fluxes, both directly through multiple reflection of SW radiation as well as indirectly through modifying the atmospheric state above these surface types.

**Sun position and TOA insolation**

SW↓ radiation at the surface exhibits strong variations with sun position (Hottel, 1976; Curry et al., 1996). Sun position directly determines the amount of SW insolation, but also affects atmospheric SW transmittance. Sun position is a function of time and location, and the representativeness of CloudSat/CALIPSO SW↓ retrievals therefore depends heavily on the difference in sun position between satellite footprint and the POI.

Furthermore, CloudSat and CALIPSO cross the equator at around 1:30 pm solar time on the day side of the earth, and again around 1:30 am solar time on the night side. The implications of such fixed overpass times are a non-representative sampling of sun position and TOA insolation with respect to the full diurnal cycle observed at the POI.

**Altitude**

In case of two nearby locations with similar atmospheric conditions but at different altitudes, downwelling radiation at the surface strongly varies with the difference in their altitudes. LW↓ radiation is determined by the atmospheric temperature and emissivity. Under clear-sky conditions, the latter is mainly a function of the atmospheric water vapour (Rodgers, 1967), whereas under cloudy conditions it is largely determined by the amount of cloud liquid and ice water in the atmospheric column (Shupe and Intrieri, 2004). Under similar atmospheric conditions at nearby locations, LW radiation differences are mainly explained by temperature differences that emerge from altitude variations through the atmospheric lapse rate, and related humidity variations.

SW↓ radiation is determined by solar insolation at the TOA and the atmospheric SW transmittance. At nearby locations with different altitudes, the shorter atmospheric path that is associated with the higher altitude leads to a higher transmittance compared to the longer atmospheric path that is associated with the lower altitude. This is explained by the absolute air mass between the source of solar radiation and the surface (Laue, 1970). Radiative flux retrievals, both LW↓ and SW↓, at nearby locations therefore strongly depend on altitude differences between these locations.



## 3.2 Temporal representativeness

The spatial pattern of CloudSat/CALIPSO overpasses is shown for the Arctic in Fig. 3a. The sampling rate is determined by the amount of overpasses within a given timeframe. At a specified POI, this rate increases with the maximum allowed distance from this POI for which a satellite overpass is still considered representative (blue circles in Fig. 3a). In addition, the sampling rate

of the near-polar orbiting CloudSat/CALIPSO satellites increases towards the poles, up to a maximum of $82°$ beyond which there is no longer satellite coverage. The average time between subsequent overpasses in function of latitude and maximum allowed distance is shown in Fig. 3b. Sampling frequency by CloudSat/CALIPSO varies between only once every almost 10 days at latitudes of about $60°$ and maximum allowed distances <50 km, to almost 10 times a day at latitudes towards $80°$ and maximum allowed distances up to 1000 km.

The best estimates of radiative fluxes are provided by the largest amount of CloudSat/CALIPSO radiative flux samples. Hence, sampling frequency is an important factor to consider when using satellite observations for radiative flux retrievals. This concept is illustrated for monthly means in Fig. 4, where AWS flux observations on hourly timescales were sampled at a progressively coarser temporal resolution. Monthly mean radiative fluxes were calculated based on each subsample and compared to the monthly means calculated from the complete dataset. The results were averaged over all six AWSs, while the

range for the individual stations is shown by the shaded areas. From this analysis, it is clear that with decreasing sampling rate, the monthly mean root-mean-square error (RMSE) increases.

## 4 Methodology: smart sampling approach

To cope with the challenges related to narrow-swath satellite sampling of retrieved downwelling surface radiative fluxes, a smart sampling approach is developed in this section. The main goal of the smart sampling approach is to maximize the

sampling frequency while at the same time maximizing the representativeness of the satellite retrievals for a POI. The entire smart sampling approach is schematically shown in Fig 5, with each step explained below. The entire procedure is designed in such way that it only relies on information that is readily available from the 2B-FLXHR-LIDAR product. This approach ensures that the method can be applied to any land-based location in polar regions without the need for auxiliary information.

### 4.1 Spatial correction

The purpose of the spatial correction procedure is to select the satellite-retrieved radiative fluxes over similar surfaces and further correct them for SZA and altitude differences with respect to the POI. It consists of five main parts (schematically shown on the left side of Fig. 5): ocean and albedo masking, calculating SW transmittance, SZA correction on the SW transmittance, altitude correction on the SW transmittance and LW↓ radiation and recalculating SW↓ radiation at the POI.



### 1) Ocean and albedo masking

Since this study focuses on retrievals over land, the correction starts with a masking of CloudSat/CALIPSO observations over ocean. Moreover, we exclude the tracks over regions where the mean surface albedo in a 2° by 1° gridbox differs more than 20 % from the surface albedo around the POI, which allows for slightly different surface conditions while at the same time avoiding for example regions that are covered by bare rock while the POI is covered by snow/ice.

### 2) Calculating SW transmittance

The original surface SW↓ radiative fluxes from the CloudSat/CALIPSO satellites ($SW\downarrow_{surf,sat}$) are first used to calculate their respective SW slant path transmittances ($\tau_{sat}$) based on the instantaneous TOA SW insolation at the satellite location ($SW\downarrow_{toa,sat}$), as described by Eq. (1) (Bintanja, 1996):

$$\tau_{sat} = \frac{SW\downarrow_{surf,sat}}{SW\downarrow_{toa,sat}} \tag{1}$$

A minimum amount of SW insolation is required for a reliable retrieval of SW transmittance. Hence, a minimum threshold of 100 W m⁻² was used here to distinguish between daytime and nighttime satellite overpasses, where only daytime overpasses can be used for the transmittance calculations. Instead of removing all SW↓ samples with TOA insolation below this threshold, SW↓ surface radiative fluxes below 15 W m⁻² are retained without performing additional corrections to avoid a significant wintertime gap. Given the very small SW↓ values, this does not impact the accuracy of the retrievals.

### 3) SZA correction

Next, a correction is required to rescale the satellite-retrieved transmittance to a transmittance that would be observed at the POI under a different SZA. Equation (2) describes the relationship between the satellite-retrieved slant path transmittance of a profile $\tau_{sat}$ under a SZA $\theta_{sat}$ and the vertical transmittance $\tau_{\perp}$, at the time of overpass (Kidder and Vonder Haar, 1995).

$$\tau_{sat} = \tau_{\perp}^{(\cos\theta_{sat})^{-1}} \tag{2}$$

The corresponding slant path transmittance at the POI under a different SZA $\theta_{poi,i}$, at any time $i$, is described in a similar way by Eq. (3):

$$\tau_{poi,i} = \tau_{\perp}^{(\cos\theta_{poi,i})^{-1}} \tag{3}$$



By combining Eq. (2) and Eq. (3) and under the assumption that the atmospheric composition over the POI is similar to the satellite profile, a corrected SW transmittance at the POI at time $i$ follows from the satellite-retrieved transmittance and their respective SZAs:

$$\tau_{poi,i} = \tau_{sat}^{[\cos\theta_{sat}(\cos\theta_{poi,i})^{-1}]}$$ (4)

### 4) Altitude correction

To quantify the effect of altitude differences between the satellite footprint and the POI, we binned all available satellite-retrieved LW↓ fluxes and SW↓ transmittances from 2007-2010 within 1000 km of each of the six AWS locations according to surface altitude of the satellite footprints, information that is available in the 2B-FLXHR-LIDAR product. This yields for each individual AWS location unique relationships between surface altitude and mean LW↓ fluxes and SW↓ transmittances, as shown in Fig. 6. LW↓ radiation exhibits a strong linear correlation with mean altitude, related to the approximately linear temperature lapse rate and related humidity profiles. SW transmittance in turn shows a slightly different relationship with altitude, and best overall fit was attained using a power relation. Such relation can be explained by the decreasing absolute air mass of the atmospheric column above the surface with altitude (Laue, 1970) and decreases in water vapor and aerosol amounts that strongly contribute to the volume extinction coefficient for SW radiation (Ramaswamy and Freidenreich, 1991; Henzing et al., 2004).

Based on this altitude relationship from 2B-FLXHR-LIDAR profiles, the satellite LW↓ radiation retrievals are rescaled to the corresponding LW↓ that are expected at the POI based on the difference in altitude, using the derived unique linear relationship for each location, as shown in Fig. 6 for the six AWS locations. The SW transmittance at an altitude of each satellite footprint is rescaled to SW transmittance that is expected at the altitude of the POI, based on the difference in altitude and the unique power relationship derived from the available retrievals at each location (Fig. 6). Since no auxiliary information was required to derive these relationships for the individual locations, new relationships can readily be calculated for any new POI in polar regions.

### 5) Calculating SW↓$_{poi,i}$

Finally, the SW transmittance at the POI at time $i$ which is corrected for SZA and altitude differences, is converted back to the corresponding SW↓ radiation at the POI at time $i$, using the instantaneous TOA SW insolation:

$$\text{SW}\downarrow_{surf,poi,i} = \tau_{poi,i} \, \text{SW}\downarrow_{toa,i}$$ (5)

Figure 7 illustrates the effect of the spatial correction procedure for the example of the PE station in Antarctica. Comparison of yearly mean biases in 2° by 1° gridboxes with respect to the satellite retrievals near the POI before and after spatial correction





clearly shows a strong increase in spatial representativeness. Remaining differences are related to other factors, such as varying cloud regimes.

## 4.2 Optimized sampling

As indicated on the right-hand side of Fig. 5, a maximum distance can now be iteratively determined for each location that is needed to reach a desired sampling frequency. From the corresponding maximum distance to reach that sampling frequency, a final dataset with representative SW↓ and LW↓ retrievals is constructed, that can be used to calculate statistical properties and uncertainties on surface radiative fluxes.

Due to the fixed overpass times of CloudSat/CALIPSO and consequently non-diurnal estimates of SW radiation, the final step involves simulating the diurnal cycle for SW↓ radiation. This is done by retrieving the $SW\downarrow_{surf,poi,i}$ in Eq. (5) for every hour, and then calculating the average to yield the diurnal-weighted $SW\downarrow_{poi,dw}$:

$$SW\downarrow_{poi,dw} = \frac{\sum\limits_{i=1}^{24} \tau_{poi,i}\, SW\downarrow_{toa,i}}{24} \tag{6}$$

## 4.3 Uncertainty retrievals

The two main sources of uncertainty in the final CloudSat/CALIPSO SW↓ and LW↓ datasets arise from remaining lack of representativeness in function of distance between the samples and the POI ($\epsilon_{dist}$) , and from a limited sampling frequency ($\epsilon_{sf}$). After the spatial correction procedure, the filtered subset of satellite observations only contains profiles over similar surfaces that are corrected for SZA and altitude differences with respect to the POI. Remaining differences in radiative fluxes in function of distance from the POI are due to progressively more different factors such as cloud regimes, temperature and humidity profiles and others, that decrease the agreement between the location where the satellite radiative fluxes are retrieved and the radiative fluxes that would be observed at the POI, thereby enhancing the uncertainty on the retrievals.

We quantified this uncertainty by comparing the radiative fluxes at a specified distance from the POI to the satellite-retrieved radiative flux at the POI itself, which is possible for all available satellite tracks that pass within 50 km of the POI which is considered here as a reasonably close overpass. The result shown in Fig. 8 demonstrates that for both LW↓ and SW↓ radiation the uncertainty in terms of RMSE increases progressively with distance, although the rate of this increase varies considerably between the locations, related to the (in)homogeneity of the regions around the POI. The consequence is that including more retrievals at a larger distance inevitably increases the uncertainty related to representativeness issues. Fig. 8 provides a means of estimating these uncertainties for the radiative flux retrievals in function of distance ($\epsilon_{dist}$).

In addition, a higher sampling frequency leads to a lower sampling uncertainty ($\epsilon_{sf}$) and vice versa. The sampling error $\epsilon_{sf}$ is calculated based on the relationship between sampling frequency and the average RMSE derived from AWS measurements, as shown for the example of monthly means in Fig. 4. The final dataset therefore has two main sources of uncertainty, related





to the limited sampling frequency ($\epsilon_{sf}$) and to the distance between the samples and the POI ($\epsilon_{dist}$). Assuming that these two sources are independent, the total uncertainty $\epsilon_{tot}$ is described by Eq. (7).

$$\epsilon_{tot} = \sqrt{(\epsilon_{sf})^2 + (\epsilon_{dist})^2} \tag{7}$$

## 5 Results

The desired sampling frequency of the smart sampling approach was iteratively determined based on both the agreement with ground-based measurements and the magnitude of the total uncertainty $\epsilon_{tot}$. A daily sampling frequency was found to yield optimal results. Higher frequencies require to sample at larger distances from the POI which increases the distance-related uncertainty $\epsilon_{dist}$. Lower frequencies increase the sampling-related uncertainty $\epsilon_{sf}$. In terms of comparisons with measured radiative fluxes at the AWSs as well, a daily frequency yields best agreements.

The performance of the smart sampling approach is compared to a limited-distance sampling technique, defined here as all uncorrected satellite samples within 10 km from the POI, the average maximum distance to the closest satellite overpass on any given location at 70° latitude. Each unique satellite track is considered an overpass, whereas one satellite profile in an overpass is considered to be a sample.

The monthly number of available CloudSat/CALIPSO overpasses and samples for both sampling <10 km from the POI and

smart sampling is shown in Table 2. The amount of CloudSat/CALIPSO overpasses is on average only twice per month for limited-distance sampling, which increases to 35 times per month for the smart sampling approach. This is slightly more than a daily overpass, which was set here as the desired sampling frequency. The average total amount of monthly CloudSat/CALIPSO samples increases from 33 to 8,412 (LW↓) and from 33 to 7,973 (SW↓), showing the strong increase in sampling frequency for the smart sampling approach as opposed to sampling <10 km from the POI.

For the comparison between sampling techniques, we calculated statistical properties on monthly samples, since few or no samples are available on finer temporal resolutions for the limited-distance sampling technique. In addition, monthly timescales are often the temporal resolution of end-products, such as the Level-3 CloudSat products. Compared to the limited-distance situation, the smart sampling approach clearly yields better results, both for the LW↓ radiation (Fig. 9) and the SW↓ radiation (Fig. 10). Overall, agreement in terms of bias and RMSE has significantly increased for LW↓ radiation (Table 3), with an

average monthly mean bias reduction from 6 W m$^{-2}$ to 2 W m$^{-2}$ and a RMSE decrease of 23 W m$^{-2}$ to 10 W m$^{-2}$. Regarding SW↓ radiation, the improvement is mostly found in a strongly decreased RMSE from 43 W m$^{-2}$ to 14 W m$^{-2}$, with little effect on the bias.

These significant improvements are mainly the result of greatly increased sampling frequencies (Table 2) with simultaneously enhanced spatial representativeness after the spatial correction procedure. The decrease in RMSE from sampling <10 km

from the POI to smart sampling becomes smaller on coarser temporal resolutions such as yearly values, especially for the LW↓ fluxes (not shown). This indicates that the added value of the smart sampling approach is largest on finer temporal resolutions, where the limited-distance sampling technique suffers from severely limited sampling frequencies.



One location that stands out with a worse agreement in SW↓ fluxes is the NYA station, where SW↓ fluxes are significantly overestimated in the satellite data. Upon closer investigation, this is caused by much higher summer surface albedo values used in the 2B-FLXHR-LIDAR algorithm (∼0.75) as opposed to what is observed at the AWS station where albedo can decrease down to ∼0.15. This is a limitation in the 2B-FLXHR-LIDAR dataset, where coastal regions or regions that have prolonged melt events might be characterized by albedo values that are too high in the satellite dataset (Kay and L'Ecuyer, 2013), with biases in the SW↓ fluxes as a consequence.

In addition to monthly mean radiative fluxes, the increased sampling frequency of the smart sampling approach further leads to a greater coverage of intra-monthly radiative flux values, as illustrated by comparing the 10[th] percentile (P10) and 90[th] percentile (P90) LW↓ and SW↓ values from 2B-FLXHR-LIDAR against the observations from the AWSs (Fig. 11 and Fig. 12). The agreement with AWS observations is much higher for the smart sampling approach, although the P10 for SW↓ fluxes clearly shows an overestimation. This overestimation suggests high biases for low SW transmittance values, which can be explained by the minimum threshold of 100 W m$^{-2}$ of TOA insolation that was set to calculate the SW transmittance, while transmittance is known to be lower for lower insolation values (Young, 1994).

Remaining differences between satellite-retrieved fluxes and AWS observations that are beyond the included uncertainty estimates can be attributed to issues not taken into account in the spatial correction procedure. For example, the persistent overestimation in LW↓ radiation at Dome-C over the Antarctic plateau is likely related to a warm bias in ERA-Interim (Fréville et al., 2014; Jones and Lister, 2014) which provides the temperature profiles for the flux calculations in 2B-FLXHR-LIDAR. Furthermore, also the AWS observations contain measurement uncertainties, and these stations can also be located in very specific environments that are difficult to capture by satellite remote sensing. Despite these limitations, the smart sampling approach yields very good agreements with observations at the polar land sites, demonstrating both the good performance of the smart sampling approach, as well as the inherent quality of the CloudSat/CALIPSO retrieved radiative fluxes.

We also compared the results from the smart sampling approach against SW↓ and LW↓ fluxes from ERA-Interim reanalyses (Dee et al., 2011) in Table 4. In general, the satellite retrievals outperform ERA-Interim for LW↓ fluxes, although this depends on the station. At the same time, ERA-Interim performs slightly better than the satellite retrievals for SW↓ fluxes. This suggests that including active satellite cloud observations is especially beneficial for the retrieval of LW↓ fluxes, while an explicitly simulated full diurnal cycle of SW↓ radiation in reanalysis data such as ERA-Interim enhances the agreement with AWS observations at most locations. Moreover, since most of the AWS locations considered here are located near the coast, the smart sampling approach is forced to sample the satellite data more inland. Both atmospheric and surface conditions can therefore be significantly different from the conditions at the AWS stations themselves. This is especially important for surface albedo values that tend to be higher in the satellite samples taken further inland with consequent overestimations in the SW↓ fluxes.



## 6 Application

The estimation of downwelling surface radiative fluxes for any given location on land in the polar regions exclusively using 2B-FLXHR-LIDAR data, provides useful applications. This is particularly interesting for locations where no or few ground observations are available. As an example, two locations are explored for which there are few ground observations available (blue dots in Fig. 2). Pine Island Glacier in Antarctica is one of the fastest melting glaciers on the continent with its retreat accelerating rapidly (Jenkins et al., 2010), although observations of the energy budget are scarce. In the Arctic, over Greenland, most of the AWSs are situated near the coast with numerous large blind spots in the interior of the ice sheet, where surface melt was reported in the July 2012 extreme melt event (Nghiem et al., 2012). Therefore, we demonstrate the smart sampling approach for Pine Island Glacier ('PIG', 75.17° S, 100° W) and the Greenland northern interior ('GRINT', 77° N, 42° E).

To include the enhanced representation of intra-monthly variability in radiative fluxes, we calculated 5-day moving averages over the entire final SW↓ and LW↓ datasets that result from the smart sampling approach (Fig 5), and compared it to what would be available from limited-distance sampling of satellite observations <10 km from these two locations. In order to verify that the resulting 5-day moving averages are representative for what is observed on the ground, we repeated this exercise for the Georg von Neumayer (GVN) station in Antarctica, where the results are compared to AWS observations (Fig. 13).

The results clearly show the added value of the smart sampling approach with strongly increased sampling frequencies that significantly reduce the amount of missing data when compared to the limited-distance sampling method. Apart from a reduction in data gaps, also the agreement with respect to AWS observations at GVN is enhanced by the smart sampling approach, suggesting that also the retrievals at PIG and GRINT will be more representative for those locations as opposed to what is retrieved by limited-distance sampling <10 km from the locations. Remaining data gaps in the smart sampling approach are due to missing 2B-FLXHR-LIDAR data in the event that one or more algorithm inputs were not available.

## 7 Discussion

Observations of surface radiative fluxes in polar regions are crucial, both in terms of increased understanding of the SEB (e.g. van den Broeke, 2004; Sedlar et al., 2011; Gorodetskaya et al., 2015), as well as for evaluation purposes of climate models (e.g. Gallée and Gorodetskaya, 2010; King et al., 2015; English et al., 2015). The methodology developed here can significantly increase the amount of satellite-based retrievals of SW↓ and LW↓ radiation on a monthly basis, or even at finer temporal resolutions as shown in Fig. 13. While we performed SZA correction for a simulation of the diurnal cycle, a Level-3 monthly, gridded version of the CloudSat radiative fluxes and heating rates product that incorporates an explicit diurnal correction will be made available as part of the upcoming Release 05 of the dataset. On timescales shorter than a month, however, our SZA correction provides an efficient method to simulate the diurnal-weighted SW↓ fluxes.

For capturing real diurnal variations, however, the smart sampling approach is insufficient due to the limited amount of overpasses and the much higher uncertainties on the individual satellite profiles. Nevertheless, in such cases these satellite retrievals may be used in a hybrid approach where satellite observations and climate model data are combined to yield best estimates of diurnal surface radiative fluxes, as shown in Van Tricht et al. (2016).





This study has focused on downwelling radiative fluxes, while upwelling radiative fluxes are equally important. However, LW↑ fluxes from the surface are exclusively a function of surface skin temperature and emissivity which are taken from ERA-Interim reanalyses and IGBP data in the 2B-FLXHR-LIDAR algorithm (Henderson et al., 2013), meaning that Cloud-Sat/CALIPSO observations do not provide added value for estimating LW↑ fluxes at the surface. SW↑ fluxes at the surface are

determined by the surface albedo value. Since the 2B-FLXHR-LIDAR algorithm relies on external information for the surface albedo values from IGBP data with related spatial and temporal resolutions that do not always closely agree with observations on the ground (Kay and L'Ecuyer, 2013), SW↑ radiative fluxes were not included here.

In addition, the complete SEB contains turbulent fluxes as well, which can play an important role in energy exchanges between surface and atmosphere (Curry et al., 2000; Van Den Broeke et al., 2006; de Boer et al., 2014) and in mass-related

processes such as sublimation (Thiery et al., 2012), in addition to the radiative fluxes considered here. Since turbulent fluxes cannot be retrieved from CloudSat/CALIPSO observations, these are not included in the present study. For a complete insight in the SEB, other information sources therefore need to be addressed to include turbulent heat fluxes in the analyses as well.

Lastly, this study has mainly focused on developing a methodology to retrieve SW↓ and LW↓ radiative fluxes at discrete land-based locations in polar regions. Yet, the smart sampling approach can in principle be used for large-scale applications as

well. While for such applications gridded datasets are mostly used, the smart sampling approach can contribute to enhancing the spatial and temporal resolution of a gridded version of the 2B-FLXHR-LIDAR product. Altough extending the smart sampling approach for such large-scale applications was beyond the scope of this study, it will be an important subject of future work.

## 8  Conclusions

In this study, we demonstrated a methodology to optimally sample narrow-swath satellite-based radiative flux retrievals for

estimating downwelling long-wave (LW↓) and short-wave (SW↓) fluxes at any given point-of-interest (POI) on land in the polar regions below 82° latitude. Increasing the distance between the satellite observations and the POI leads to a tradeoff, where sampling frequency is enhanced, but spatial representativeness is reduced.

This decrease in spatial representativeness can be mitigated to some degree by implementing a smart sampling approach. It is shown here that a spatial correction procedure can significantly improve the spatial representativeness of satellite retrievals.

This includes (1) ocean and albedo masking, (2) conversion from SW↓ radiation at the surface to SW transmittance, (3) solar zenith angle correction on transmittance values, (4) altitude correction on SW transmittance and LW↓ fluxes and (5) conversion of corrected SW transmittances back to SW↓ fluxes. Optimized sampling then comprises the construction of a final SW↓ and LW↓ fluxes dataset, where for SW↓ fluxes the diurnal cycle is simulated. This is done in an iterative way of increasing the distance to the POI until a desired sampling frequency is reached. A daily frequency was determined here to yield optimal

results.

Implementing the smart sampling approach is shown to increase on average the amount of unique satellite overpasses from only twice each month for limited-distance sampling <10 km from the POI to 35 each month, with a consequent increase in total amount of satellite samples from 33 to 8,412 (LW↓) and 7,973 (SW↓). The enhanced agreement with AWS observations



is illustrated on monthly samples with reduced root-mean-square errors from 23 W m$^{-2}$ to 10 W m$^{-2}$ (LW↓) and 43 W m$^{-2}$ to 14 W m$^{-2}$ (SW↓), in addition to a significantly better representation of intra-monthly variation. It is found that the improvement by using the smart sampling approach is largest on finer temporal resolutions, since the limited-distance sampling technique <10 km from the POI has very limited sampling frequencies at these timescales. The smart sampling approach is finally

applied to Pine Island Glacier and the Greenland northern interior, regions of scientific interest where few or no ground-based observations are available. The smart sampling approach is able to estimate 5-day moving averages of both LW↓ and SW↓ radiative fluxes for these locations.

Overall, we conclude that the developed smart sampling approach allows to retrieve downwelling surface radiative fluxes at any given location over land in the polar regions, where the calculated uncertainties indicate how well CloudSat and CALIPSO

are able to estimate these radiative fluxes. Homogenous regions with good satellite coverage result in high confidence of the retrieved radiative fluxes, while heterogenous regions with limited satellite coverage result in lower confidence. These results may help reducing the observational gaps that remain in polar regions. By filling these gaps and enhancing the temporal resolution, the described smart sampling approach can provide data that we need to improve our understanding of the polar surface energy budget.

**Data and code availability**

The monthly means, 5-day moving average time series and smart sampling code can be made available upon request.

*Acknowledgements.* K.V.T. and S.L. are funded by the Research Foundation Flanders (FWO). I.V.G. was supported via the project HY-DRANT funded by the Belgian Science Policy Office under grant number EN/01/4B. This work is further supported by the Belgian Federal Science Policy Office project AEROCLOUD (BR/143/A2). We are sincerely thankful to all scientists who are responsible for high-quality

data acquisition at the various BSRN sites. BSRN data used in this study are available at http://bsrn.awi.de/en/data/. We further thank Wim Boot, Carleen Reijmer and Michiel van den Broeke (Institute for Marine and Atmospheric research Utrecht, the Netherlands) for the PE AWS development, technical support and raw data processing. The CloudSat Level-2 Fluxes and Heating Rates product can be acquired through the CloudSat data processing center at http://www.cloudsat.cira.colostate.edu.



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





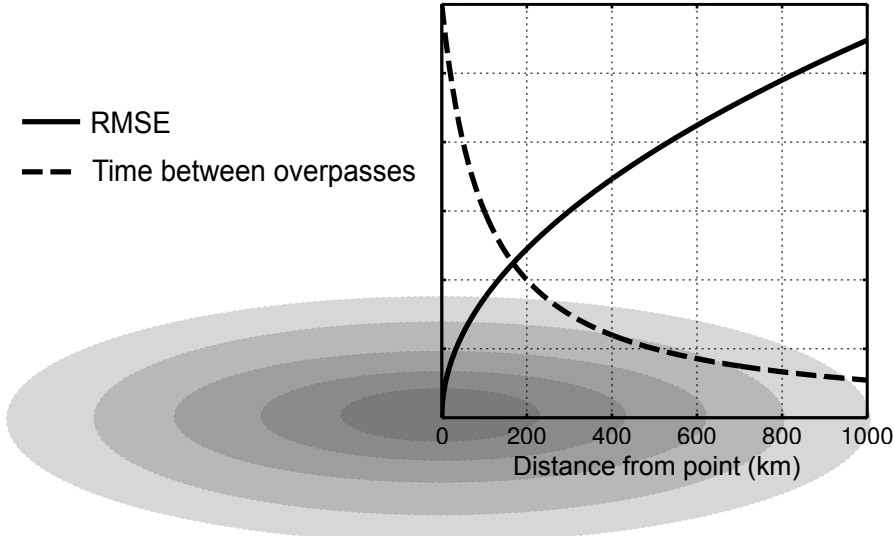

**Figure 1.** Conceptual illustration of tradeoff between sampling frequency and RMSE with increasing distance from a location. The time between subsequent satellite overpasses decreases with distance, but the agreement between these overpasses and the conditions at the location decreases as well.

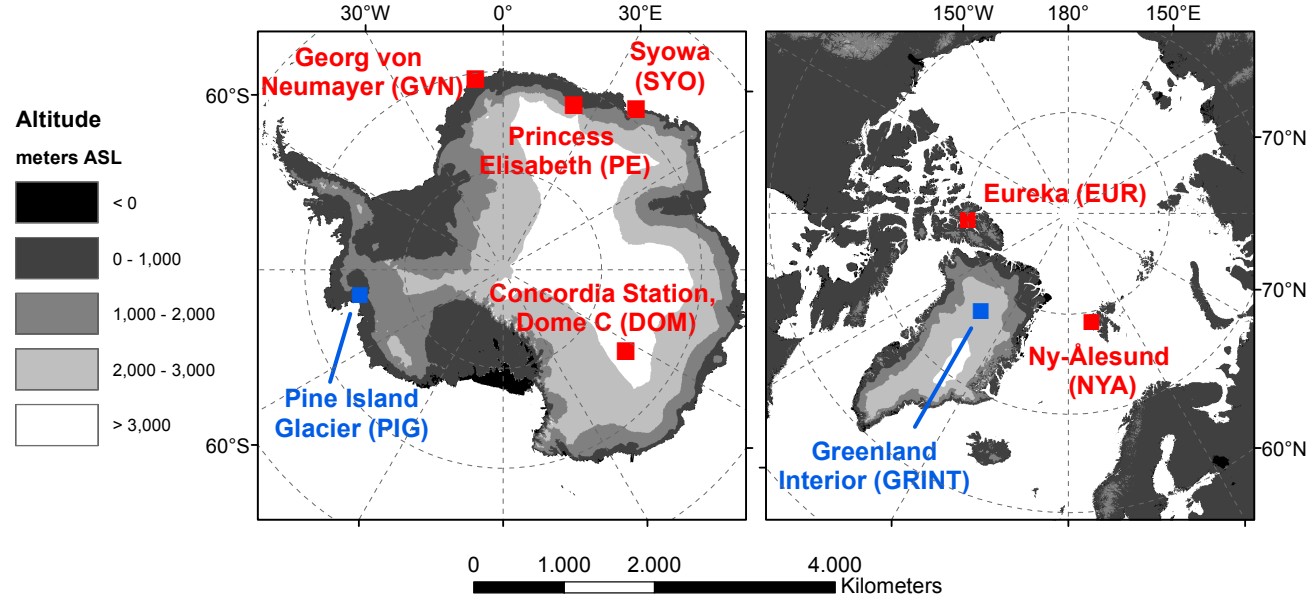

**Figure 2.** Locations of the six AWSs (red) and two new locations (blue).





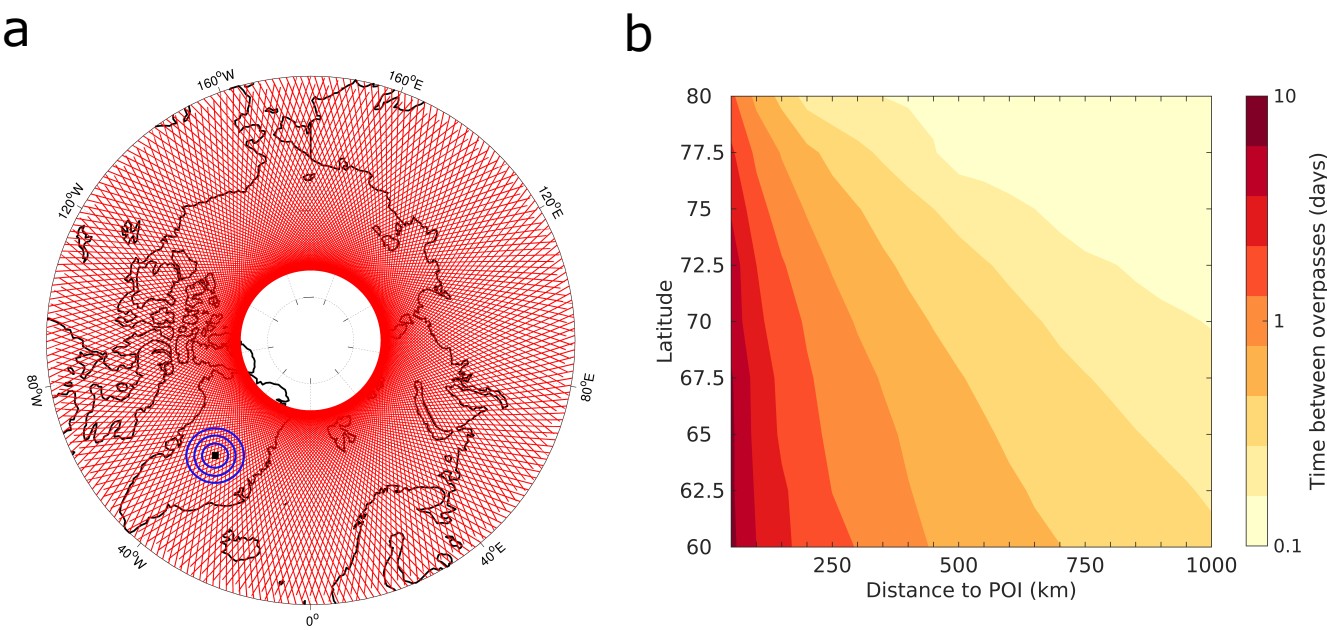

**Figure 3. (a)** CloudSat/CALIPSO overpass tracks in the Arctic for one repeat cycle of 16 days. The blue circles show the increased sampling rate when a larger area is taken into account. **(b)** Maximum CloudSat/CALIPSO sampling frequency in function of both distance to the POI and latitude. It should be noted that this is the theoretical maximum sampling frequency at each location. If satellite samples are excluded in processing steps, the real sampling frequency decreases.





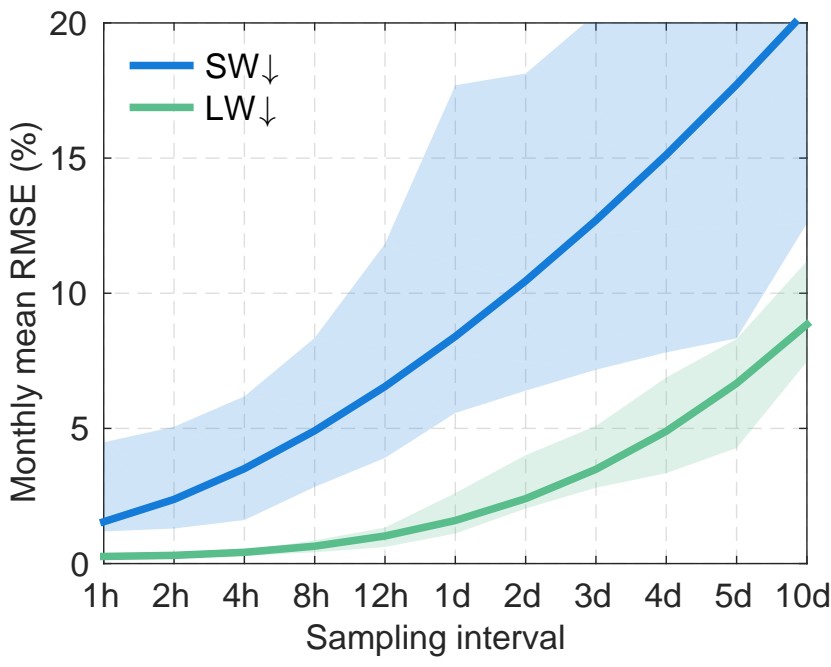

**Figure 4.** Monthly mean SW↓ and LW↓ RMSE (%) in function of sampling interval as derived from six AWSs. The RMSE was calculated by comparing the monthly mean estimates based on a subsample of data with a specified sampling interval to the full hourly datasets. The two curves represent the average relationship, while the shaded areas indicate the range for the different stations.



**Figure 5.** Schematic representation of the smart sampling approach.



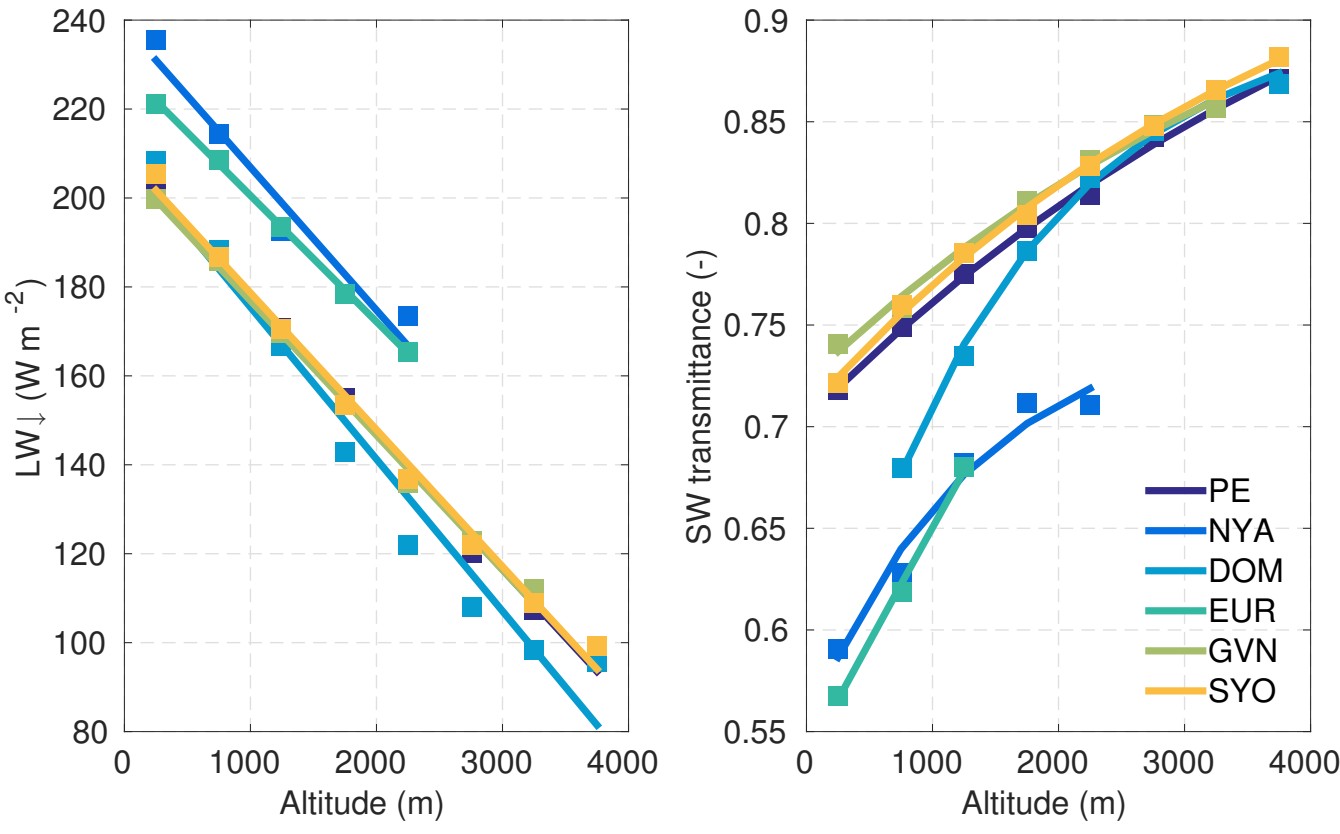

**Figure 6.** Relationship between altitude and downwelling radiative fluxes: LW↓ radiation (left) and SW↓ transmittance (right), for the six AWS locations, based on all available satellite-retrieved LW↓ fluxes and SW↓ transmittances between 2007-2010 within 1000 km of each of the six AWS locations according to surface altitude of the satellite footprints.





**Figure 7.** Monthly mean bias for each 2° by 1 ° gridbox with respect to center pixel in which the AWS is located, before and after spatial correction for the example of PE, Antarctica. These results are based on all 2B-FLXHR-LIDAR data from 2007-2010 within a distance of 1000 km from the station. It should be noted that the comparison before spatial correction here has also been masked for ocean and different surface albedos.





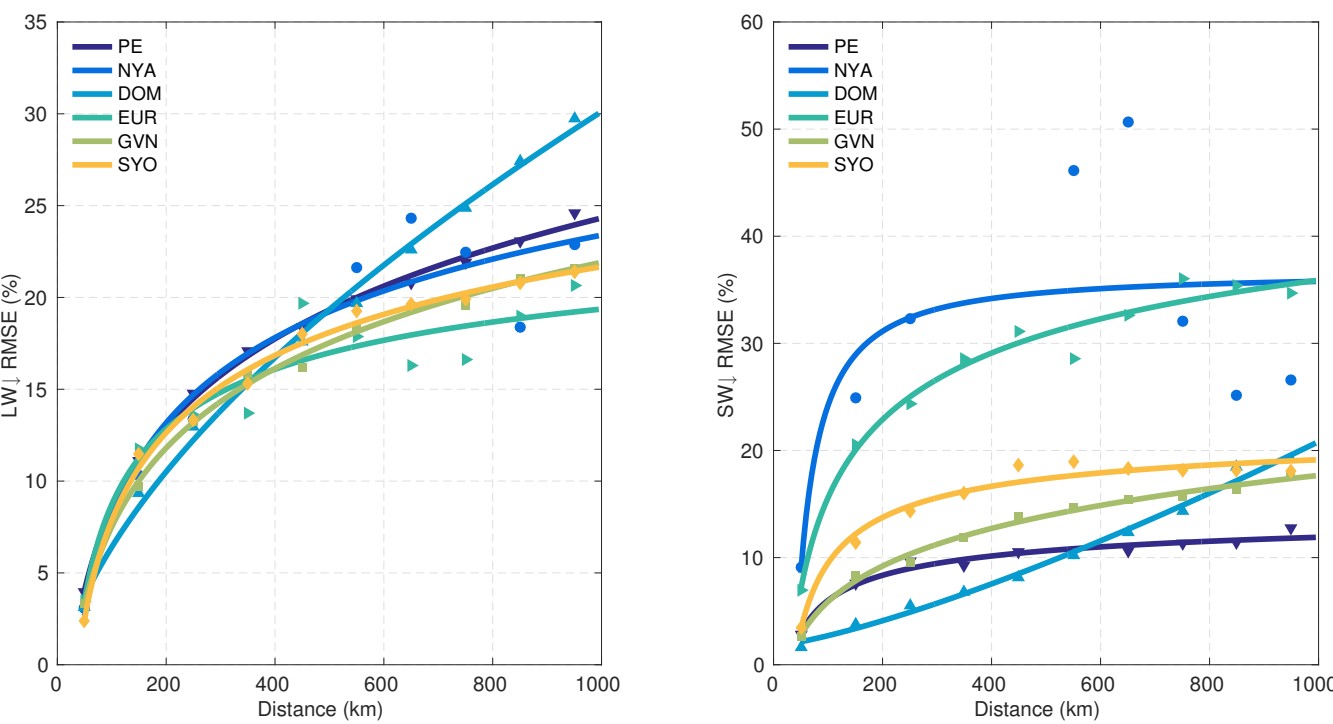

**Figure 8.** Radiative flux RMSE (%) in function of distance to the POI for six AWS stations, based exclusively on 2B-FLXHR-LIDAR data (2007-2010). The RMSE is calculated based on all satellite tracks that pass within 50 km of the POI, where the retrieved radiative fluxes at a certain distance were compared to the retrieved fluxes within these 50 km from the POI.



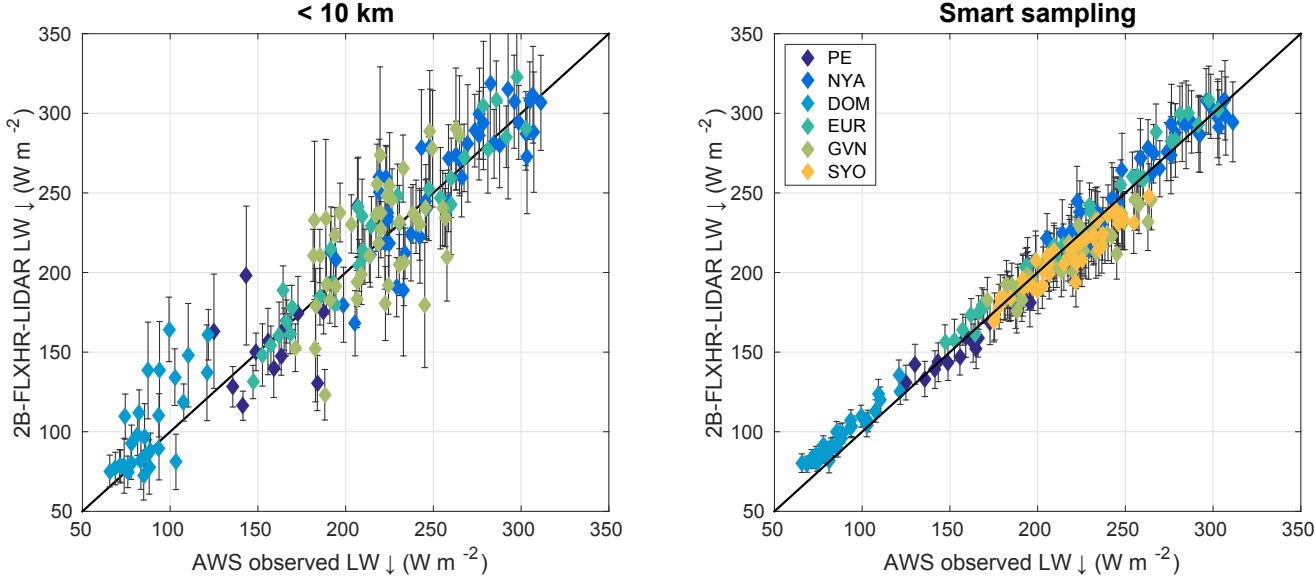

**Figure 9.** Monthly mean LW↓ radiation comparison between 2B-FLXHR-LIDAR and AWS (2007-2010). (left) Based on retrievals comprising of all satellite samples <10 km from station. (right) Based on all satellite samples resulting from the smart sampling approach.

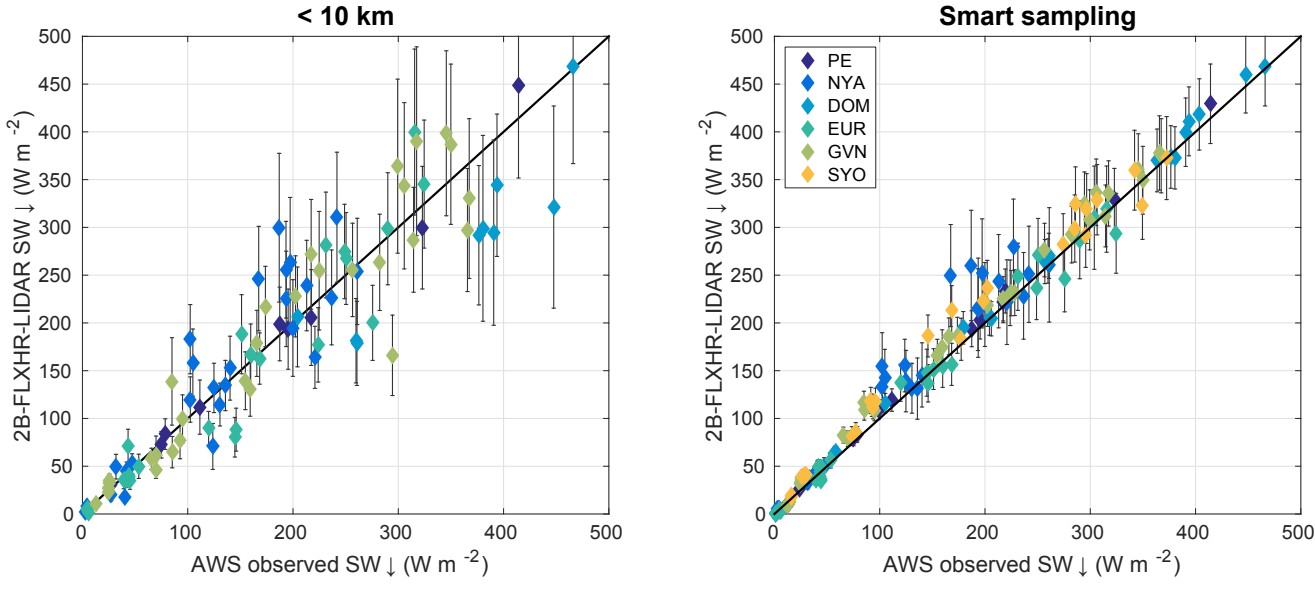

**Figure 10.** Monthly mean SW↓ radiation comparison between 2B-FLXHR-LIDAR and AWS (2007-2010). (left) Based on retrievals comprising of all satellite samples <10 km from station. (right) Based on all satellite samples resulting from the smart sampling approach.







**Figure 11.** Monthly mean LW↓ radiation comparison between 2B-FLXHR-LIDAR and AWS (2007-2010). (upper left) Based on retrievals comprising of all satellite samples <10 km from station, **P10**. (upper right) Smart sampling approach, **P10**. (lower left) Retrievals <10 km from station, **P90**. (lower right) Based on all satellite samples resulting from the smart sampling approach, **P90**.







**Figure 12.** Monthly mean SW↓ radiation comparison between 2B-FLXHR-LIDAR and AWS (2007-2010). (upper left) Based on retrievals comprising of all satellite samples <10 km from station, **P10**. (upper right) Smart sampling approach, **P10**. (lower left) Retrievals <10 km from station, **P90**. (lower right) Based on all satellite samples resulting from the smart sampling approach, **P90**.





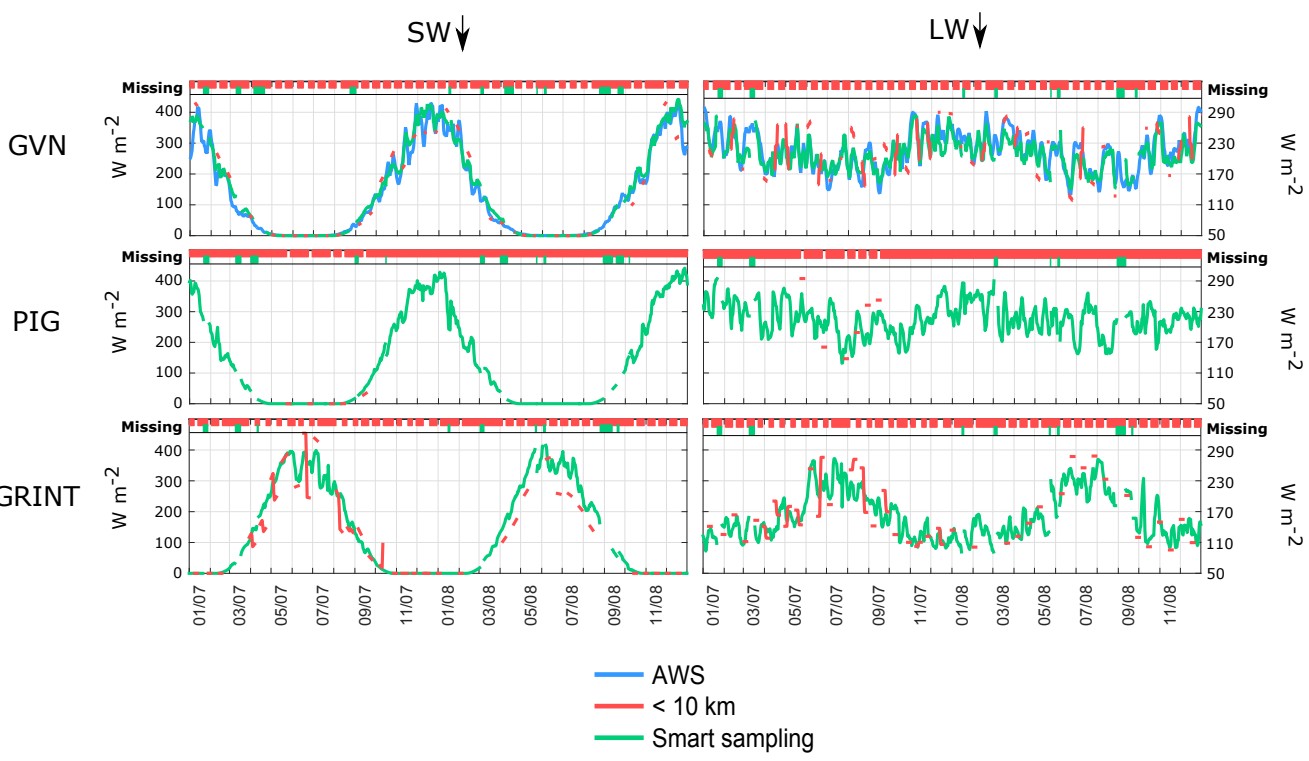

**Figure 13.** 5-day moving average SW↓ and LW↓ fluxes for GVN, PIG and GRINT (January 2007 - December 2008). The available AWS observations at GVN are shown in blue. The limited-distance sampling <10 km from the POI (red) shows significantly more data gaps compared to the smart sampling approach (green), while also the agreement with AWS observations is better for the smart sampling approach.





**Table 1.** Description of the location and instrument specifications of the six AWSs: Eureka (EUR), Ny-Alesund (NYA), Georg von Neumayer (GVN), Concordia Station Dome C (DOM), Princess Elisabeth (PE), and Syowa (SYO). Measurement accuracies are as reported by the manufacturer on daily totals.

| Station | EUR | NYA | GVN | DOM | PE | SYO |
|---|---|---|---|---|---|---|
| Latitude | 79.98 | 78.93 | -70.65 | -75.10 | -71.95 | -69.01 |
| Longitude | -85.93 | 11.93 | -8.25 | 123.38 | 23.35 | 39.59 |
| Altitude (m) | 85 | 11 | 42 | 3,233 | 1,382 | 18 |
| Surface type | Tundra | Tundra | Iceshelf | Glacier | Snow | Sea ice |
| Topography type | Hilly | Mountain valley | Flat | Flat | Mountains proximity | Hilly |
| SW instrument | K&Z (CM21) | K&Z (CM11) | K&Z (CM11) | K&Z (CM22) | K&Z (CM3) | K&Z (CM21) |
| SW accuracy | 2 % | 3 % | 3 % | 2 % | 10 % | 2 % |
| LW instrument | Eppley PIR | Eppley PIR | Eppley PIR | K&Z (CG4) | K&Z (CG3) | Eppley PIR |
| LW accuracy | 5 % | 5 % | 5 % | 3 % | 10 % | 5 % |

**Table 2.** Monthly mean total number of unique CloudSat/CALIPSO overpasses and total number of individual CloudSat/CALIPSO samples for the limited-distance sampling technique <10 km from the POI (**L**) and the smart sampling approach (**S**).

| | LW↓ | | | | SW↓ | | | |
|---|---|---|---|---|---|---|---|---|
| | Overpasses | | Samples | | Overpasses | | Samples | |
| Station | L | S | L | S | L | S | L | S |
| EUR | 3 | 39 | 48 | 4,136 | 3 | 38 | 48 | 3,966 |
| NYA | 3 | 34 | 47 | 3,240 | 3 | 35 | 47 | 3,139 |
| GVN | 3 | 34 | 44 | 8,210 | 3 | 34 | 44 | 7,671 |
| DOM | 1 | 35 | 24 | 12,892 | 1 | 35 | 24 | 12,074 |
| PE | 2 | 34 | 37 | 12,020 | 2 | 34 | 37 | 11,346 |
| SYO | 0 | 34 | 0 | 9,975 | 0 | 34 | 0 | 9,641 |
| **Mean** | **2** | **35** | **33** | **8,412** | **2** | **35** | **33** | **7,973** |




**Table 3.** Statistical comparison of CloudSat/CALIPSO retrieved surface radiative fluxes in terms of bias and RMSE against ground-based AWS observations, between limited-distance sampling <10 km from the POI (**L**) and the smart sampling approach (**S**). The SYO statistics were not considered in the mean value for the smart sampling approach, since no satellite overpasses were found in the limited-distance sampling.

| | LW↓ | | | | SW↓ | | | |
| | Bias | | RMSE | | Bias | | RMSE | |
| Station | L | S | L | S | L | S | L | S |
|---|---|---|---|---|---|---|---|---|
| EUR | 5.5 | 5.9 | 14.8 | 8.5 | -2.3 | -1.0 | 35.3 | 11.3 |
| NYA | 2.4 | 3.0 | 20.8 | 9.8 | 19.0 | 16.4 | 43.4 | 28.1 |
| GVN | 3.0 | -7.1 | 29.5 | 12.4 | 1.1 | 11.6 | 41.2 | 15.0 |
| DOM | 19.2 | 9.5 | 24.8 | 9.8 | -71.2 | 5.8 | 77.7 | 8.9 |
| PE | 0.1 | -3.2 | 26.7 | 7.2 | 2.8 | 6.0 | 15.2 | 7.8 |
| SYO | N/A | (-7.1) | N/A | (10.7) | N/A | (6.4) | N/A | (21.9) |
| **Mean** | **6** | **2** | **23** | **10** | **-10** | **8** | **43** | **14** |

**Table 4.** Statistical comparison of ERA-Interim reanalyses (**ERA**) and CloudSat/CALIPSO retrieved surface radiative fluxes using the smart sampling approach (**SAT**) in terms of bias and RMSE against ground-based AWS observations.

| | LW↓ | | | | SW↓ | | | |
| | Bias | | RMSE | | Bias | | RMSE | |
| Station | ERA | SAT | ERA | SAT | ERA | SAT | ERA | SAT |
|---|---|---|---|---|---|---|---|---|
| EUR | 10.4 | 5.9 | 15.2 | 8.5 | -8.3 | -1.0 | 15.8 | 11.3 |
| NYA | -13.6 | 3.0 | 19.4 | 9.8 | -1.5 | 16.4 | 10.5 | 28.1 |
| GVN | -4.7 | -7.1 | 8.2 | 12.4 | -7.5 | 11.6 | 12.4 | 15.0 |
| DOM | 3.1 | 9.5 | 5.0 | 9.8 | -3.2 | 5.8 | 8.3 | 8.9 |
| PE | -16.4 | -3.2 | 16.8 | 7.2 | -3.8 | 6.0 | 7.8 | 7.8 |
| SYO | -1.3 | -7.1 | 10.2 | 10.7 | 1.3 | 6.4 | 11.1 | 21.9 |
| **Mean** | **-4** | **0** | **13** | **10** | **-4** | **8** | **11** | **16** |