# Peer review of "Improving satellite-retrieved surface radiative fluxes in polar regions using a smart sampling approach"

_The Cryosphere, 2016_

## Referee Comment (RC1) · Anonymous Referee #1 · 14 Jun 2016

General comments

This study presents a new method for increasing the spatio-temporal sampling of short-wave and longwave radiation retrievals from CALIPSO and CloudSat satellites. The method developed may have a strong potential for the use of active remote sensing observations over polar regions. The topic is interesting, innovative, and fits well within the scope of The Cryosphere. Moreover, the paper is well structured and well written. However, a few issues should be resolved before publication.

Specific comments

The authors have developed a method to minimize the bias relative to the spatial representativeness. There is a discussion about how the differences in surface conditions can influence the shortwave and longwave fluxes. However, I would like to see more discussion about how the differences in atmospheric conditions (cloud cover, temperature, and humidity) can induce a bias in the retrievals, and how it relates with the maximum distance chosen between the point of interest and the satellite retrieval.

Over the Greenland and the Antarctic ice sheets, there are strong gradients from coastal to inland regions in various meteorological variables such as surface temperature (Ettema et al., 2010, Fréville et al., 2014), specific humidity (Ettema et al., 2010), cloud cover (Ettema et al., 2010, Bromwich et al., 2012), and precipitation (Palerme et al., 2014). Therefore, the point of interest is probably more representative of the satellite retrieval if the direction between the point of interest and the satellite footprint is parallel to the coast than if it is perpendicular to the coast. For instance, in Antarctica, the point of interest is probably more representative of the satellite retrieval if the point of interest is oriented in an eastward or westward direction than in a southward or northward direction compared to the satellite footprint. This should be mentioned in the text. Furthermore, do the direction between the point of interest and the satellite footprint is always perpendicular to the satellite ground track in the method developed ?

Page 10, line 7. "A daily sampling frequency was found to yield optimal results". I would like to see a discussion about the maximum distance between the point of interest and the satellite retrieval for a daily sampling frequency. The information is shown in figure 3 b), but I think it should also be mentioned in this paragraph. Moreover, a curve showing the maximum distance between the CloudSat/CALIPSO footprint and the point of interest for a daily sampling frequency depending on the latitude could be added in figure 3 b).

Figure 1. What do the gray circles represent in figure 1?

Figures 9, 10, 11, and 12. The value of the correlation coefficient should be added in

all the scatter plots.

References :

Bromwich, D. H., J. P. Nicolas, K. M. Hines, J. E. Kay, E. Key, M. A. Lazzara, D. Lubin, G. M. McFarquhar, I. Gorodetskaya, D. P. Grosvenor, T. A. Lachlan-Cope, and N. van Lipzig, 2012: Tropospheric Clouds in Antarctica, Rev. Geophys., 50, RG1004, doi: 10.1029/2011RG000363.

Ettema, J., van den Broeke, M. R., van Meijgaard, E., and van de Berg, W. J.: Climate of the Greenland ice sheet using a high-resolution climate model – Part 2: Near-surface climate and energy balance, The Cryosphere, 4, 529-544, doi:10.5194/tc-4-529-2010, 2010.

Fréville, H., Brun, E., Picard, G., Tatarinova, N., Arnaud, L., Lanconelli, C., Reijmer, C., and van den Broeke, M.: Using MODIS land surface temperatures and the Crocus snow model to understand the warm bias of ERA-Interim reanalyses at the surface in Antarctica, The Cryosphere, 8, 1361-1373, doi:10.5194/tc-8-1361-2014, 2014.

Palerme, C., Kay, J. E., Genthon, C., L'Ecuyer, T., Wood, N. B., and Claud, C.: How much snow falls on the Antarctic ice sheet?, The Cryosphere, 8, 1577-1587, doi:10.5194/tc-8-1577-2014, 2014.

---

## Referee Comment (RC2) · Anonymous Referee #2 · 30 Jun 2016

Overview:

The authors present a methodology, specifically for CloudSat's 2B-FLXHR-LIDAR product, to spatially represent downward SW and LW fluxes beyond the narrow swath. The intent and methodology are clear and the paper is well written and appropriate for this journal. Due to the sparse physical observations available in the Polar Regions it is often difficult to insure the automated stations are physically representative of the whole region. While satellite products have aided this endeavor, sampling frequency still remains an issue. The methodology presented in this manuscript provides a process that can be implemented to estimate surface fluxes beyond the satellite swath to help fill gaps between satellite measurements. The authors present clear evidence that the

smart sampling improves validation statistics and increases available data points for automated stations in the Polar Regions observed. While I think this paper provides useful information that should be published there are a few subjects that should be discussed first. I have provided my comments below along with a few other suggestions.

Comments:

1) The main comment arises from understanding how the methodology handles the varying atmospheric and cloud/precipitation processes when deriving fluxes. This is particularly important when considering CloudSat observations farther from the indicated POI. A major assumption of the methodology is spatial homogeneity of atmospheric conditions - applying the transmittance (after spatial correction) to the POI. Atmospheric variables will strongly alter the transmittance and cloud location, temperature, and humidity are the largest factors in the downward flux uncertainties [as reported in Tables 6 and 7 in the cited Henderson et al (2013)]. For example, in terms of the cloud properties, it may be a useful exercise to evaluate the cloud variability as a function of distance along the CloudSat track near a POI to understand how cloud may impact the newly sampled results. Would the maximum distance needed for proper sampling frequency still be OK if cloud/atmospheric conditions were significantly different?

2) The main results in this work focus on a few POIs to demonstrate its effectiveness. While it is mentioned in the text that a large-scale extension of the sampling approach is beyond the focus of the manuscript, it would be useful to understand how practical it would be to extend this to a larger scale. I am not sure how difficult it would be to test this method on a gridded surface. It would be great if an example could be included to show how a larger sample would compare to the FLXHR-LIDAR product. If this is not possible I feel it is necessary to mention if this methodology computationally efficient to be applied at a larger scale or simple enough for a user to implement on their own.

Other Comments:

Pg 3, Line 15: I am not sure if Figure 1 is necessary. I think a description of the errors is sufficient.

Pg 3, Line 32: How similar are the broadband measurements in the AWS to the 18 bands used in FLXHR-LIDAR?

Pg 4, Line 3: Include the acronym for CPR

Pg 4, Line 8: CALIPSO/CloudSat is excellent for cloud detection and aerosol.

Pg 4, Line 9: Are you talking about 2B-FLXHR-LIDAR here? Not all CloudSat products implement auxiliary information.

Pg 4, Line 13: I think it is worth noting the high vertical resolution of the CloudSat and CALIPSO products. The vertical resolution is one feature that makes the Cloud-Sat/CALIPSO products unique in estimating radiative fluxes.

Pg 5, Line 3: Water bodies emit more radiation compared to what?

Pg5, Line 27: This difference in transmission would be only for clear sky. Are these cases taken only for clear sky measurements?

Pg 8, Line 29: The text states yearly averages, however, the caption states monthly.

Pg 9, Line 10: It would be interesting to discuss the impact including the diurnal computations have on the results.

Pg 10, Line 18: I do not like the use of the term "sampling frequency" here. While the number of samples available does increase, these are not physical measurements being made and the satellite overpass frequency is not changing.

Line 13, Line 31: Again, the number of samples is increased due to spatial processing. The number of overpasses does not change.

---

## Author Comment (AC1) · 10 Sep 2016

**Improving satellite-retrieved surface radiative fluxes in polar regions using a smart sampling approach**

K. Van Tricht, S. Lhermitte, I. Gorodetskaya, and N. van Lipzig

**Response to reviewers**
* * *
**General response**

We thank the two reviewers for the careful assessment of our work and for raising relevant concerns that need to be clarified in a revised version of the manuscript. Please find below our detailed response to the questions and comments, and our suggestions for the revised version of this manuscript. Original questions raised by the reviewers are written in *blue italic*, while answers are provided in **bold**, and suggestions for changes in the revised manuscript are written in *red italic*.

Kind regards,

K. Van Tricht and co-authors
* * *
**Response to Reviewer #1**

**General comments**

*This study presents a new method for increasing the spatio-temporal sampling of shortwave and longwave radiation retrievals from CALIPSO and CloudSat satellites. The method developed may have a strong potential for the use of active remote sensing observations over polar regions. The topic is interesting, innovative, and fits well within the scope of The Cryosphere. Moreover, the paper is well structured and well written. However, a few issues should be resolved before publication.*

**We thank the reviewer for the evaluation of our work. Below, we address point-by-point the issues that were raised.**

**Specific comments**

*RC1.1 The authors have developed a method to minimize the bias relative to the spatial representativeness. There is a discussion about how the differences in surface conditions can influence the shortwave and longwave fluxes. However, I would like to see more discussion about*

*how the differences in atmospheric conditions (cloud cover, temperature, and humidity) can induce a bias in the retrievals, and how it relates with the maximum distance chosen between the point of interest and the satellite retrieval.*

This is a very relevant point. It is true that by increasing the distance between the satellite footprint and the POI, the chances on sampling different atmospheric conditions increase as well. We would like to stress that the smart sampling approach proposed should not be used deterministically, meaning that it should indeed not be used as a tool to infer cloud conditions at the POI based on an instantaneous satellite footprint farther away. Radiative fluxes will evidently be different between locations, but this cannot be resolved without including auxiliary datasets and performing very location-specific atmospheric analyses. The main goal of this study is to develop an approach to exclusively use 2B-FLXHR-LIDAR data in a smart way to extrapolate satellite retrievals off-track. We therefore chose to develop a statistical approach, rather than a deterministic approach, which includes statistical variability in the uncertainty estimates, based on all factors that influence the radiative flux differences between locations.

This is presented in Fig. 8, where it is shown how the RMSE as a general measure of statistical variability increases with distance from the POI. This increased variability can be seen as a summary of the different factors (cloud regime, temperature and humidity profiles) that increase the radiative flux differences between the satellite footprint and the POI. We acknowledge these factors, and therefore take them into account in the uncertainty bars of Fig. 9-10.

One alternative would be to decrease the sampling distance from the POI when the atmospheric variability is higher. However, given the rather infrequent overpass of CloudSat/CALIPSO, this would result in a very specific snapshot of the atmosphere, while there is a high degree of variability. It is therefore preferential, even under these circumstances, to sample the atmospheric conditions at a larger distance from the POI, to retrieve a more average or representative state for those conditions.

We revised the manuscript as follows:

**Sect. 3.4.4 (Uncertainty retrievals):**
*After the spatial correction procedure, the filtered subset of satellite observations only contains profiles over similar surfaces that are corrected for SZA and altitude differences with respect to the POI. Remaining differences in radiative fluxes in function of distance from the POI are due to other factors. By sampling at a larger distance from the POI, chances increase that atmospheric conditions, including cloud regime and temperature and humidity profiles become significantly different from the conditions around the POI , especially at times of a highly variable atmospheric state. No correction for these factors can be applied without including auxiliary information and performing detailed atmospheric analyses, which is beyond the scope of the present study that aims at exclusively using 2B-FLXHR-LIDAR data. Yet, the uncertainty on the retrievals due to the mentioned factors can be quantified in function of distance. […]*

**Caption of Fig. 8:**
*These RMSE estimates represent the combined uncertainties in the radiative flux retrievals that arise from differences in atmospheric conditions between the satellite footprints and the POI.*

**Discussion**:
*While a correction can be performed for altitude and SZA differences, it is acknowledged that by sampling at an increasing distance from the POI, chances raise that the atmospheric conditions become significantly different from the conditions around the POI itself , especially at times of a highly*

*variable atmospheric state. It is therefore advised to not use the smart sampling approach deterministically for studying detailed cloud conditions at a specific moment in time, but rather statistically, thereby including the uncertainty estimates provided here that take into account the variability in radiative flux retrievals due to atmospheric conditions.*

*RC1.2 Over the Greenland and the Antarctic ice sheets, there are strong gradients from coastal to inland regions in various meteorological variables such as surface temperature (Ettema et al., 2010, Fréville et al., 2014), specific humidity (Ettema et al., 2010), cloud cover (Ettema et al., 2010, Bromwich et al., 2012), and precipitation (Palerme et al., 2014). Therefore, the point of interest is probably more representative of the satellite retrieval if the direction between the point of interest and the satellite footprint is parallel to the coast than if it is perpendicular to the coast. For instance, in Antarctica, the point of interest is probably more representative of the satellite retrieval if the point of interest is oriented in an eastward or westward direction than in a southward or northward direction compared to the satellite footprint. This should be mentioned in the text. Furthermore, do the direction between the point of interest and the satellite footprint is always perpendicular to the satellite ground track in the method developed ?*

**We agree that these factors can play a significant role in spatially irregular gradients, especially from coastal to more inland regions, which is also apparent in Fig. 7 (before correction) for the Princess Elisabeth station. However, for most of these gradients the spatial correction procedure strongly enhances the agreement and reduces the directionality of the gradients, also apparent in Fig. 7 (after correction). Only for the SW, some of the gradients remain, although this is mostly a latitude (and therefore sun position) effect. This and the other factors that remain after the spatial correction, are taken into account in the uncertainty estimates in function of distance (Fig. 8). To avoid further complexity of the method and to maximize the sampling frequency for a certain distance, we therefore prefer to avoid preferential directions in the sampling procedure. We suggest to add the following information (including references) to the discussion in the revised manuscript:**

*Furthermore, strong spatial gradients exist in polar regions, for example from coastal to inland regions, for surface temperature (Ettema et al., 2010, Fréville et al., 2014), specific humidity (Ettema et al., 2010), cloud cover (Ettema et al., 2010, Bromwich et al., 2012), and precipitation (Palerme et al., 2014). This leads to strong gradients in radiative fluxes, clearly seen for the example at PE in Fig. 7 before correction. Yet, the spatial correction procedure mostly resolves these issues (Fig. 7 after correction). Only for the SW radiation, a slight spatial gradient remains, but this is more a latitude and therefore sun position effect. This and the other factors that are not resolved by the spatial correction procedure, are taken into account in the uncertainty estimates of the radiative flux retrievals. Therefore, no preferential directions of sampling are determined in the smart sampling approach, to avoid additional complexity and maximize the sampling frequency at a specified distance from the POI.*

*RC1.3 Page 10, line 7. "A daily sampling frequency was found to yield optimal results". I would like to see a discussion about the maximum distance between the point of interest and the satellite retrieval for a daily sampling frequency. The information is shown in figure 3 b), but I think it should also be mentioned in this paragraph. Moreover, a curve showing the maximum distance between the CloudSat/CALIPSO footprint and the point of interest for a daily sampling frequency depending on the latitude could be added in figure 3 b).*

**We added a Table 2 that includes the maximum distances determined by the smart sampling approach for the different AWS locations, in addition to the following information in Sect. 5:**

*The maximum distance for sampling as determined by the smart sampling approach for the different AWS locations is shown in Table 2. It is clear that these numbers are higher than the theoretical distance that is needed to reach a daily sampling frequency (black dashed line in Fig 3b), due to the spatial correction procedure and exclusion of areas that are too different from the POI.*

**Furthermore, we added a black dashed line to Fig. 3b that corresponds to a daily sampling frequency, and clarified this in the caption as follows:**

*The black dashed line corresponds to an approximately daily frequency*

*RC1.3 Figure 1. What do the gray circles represent in figure 1?*

**The black circles serve to show the increasing distance from the POI, and that the maximum distance is not a fixed number. We've added this information to the caption of the figure.**

*RC1.4 Figures 9, 10, 11, and 12. The value of the correlation coefficient should be added in all the scatter plots.*

**We thank the reviewer for this good suggestion. All correlation coefficients were added to these figures.**

**Response to Reviewer #2**

**General comments**

*RC2.1 The main comment arises from understanding how the methodology handles the varying atmospheric and cloud/precipitation processes when deriving fluxes. This is particularly important when considering CloudSat observations farther from the indicated POI. A major assumption of the methodology is spatial homogeneity of atmospheric conditions - applying the transmittance (after spatial correction) to the POI. Atmospheric variables will strongly alter the transmittance and cloud location, temperature, and humidity are the largest factors in the downward flux uncertainties [as reported in Tables 6 and 7 in the cited Henderson et al (2013)]. For example, in terms of the cloud properties, it may be a useful exercise to evaluate the cloud variability as a function of distance along the CloudSat track near a POI to understand how cloud may impact the newly sampled results. Would the maximum distance needed for proper sampling frequency still be OK if cloud/atmospheric conditions were significantly different?*

Thank you for this important comment, which is in line with RC1.1. It is true that the spatial correction does not correct for differences in atmospheric conditions, since this would require auxiliary datasets and performing very location-specific atmospheric analyses. The main goal of this study is to develop an approach to exclusively use 2B-FLXHR-LIDAR data in a smart way to extrapolate satellite retrievals off-track. Indeed, radiative fluxes will be different between locations, and the degree to which this is the case depends heavily on the atmospheric variability at the times of overpass. In the example of the SW transmittance, the spatial correction takes care of altitude differences, while differences in atmospheric conditions remain untouched.

However, the suggestion of studying how much cloud variability (or atmospheric conditions in general) as a function of distance along the CloudSat track near a POI impacts the retrieved radiative fluxes, is in essence what is already shown in Fig. 8 of the original manuscript. It is shown in that figure how the RMSE as a general measure of statistical variability increases with distance from the POI. This increased variability can be seen as a summary of the different factors (cloud regime, temperature and humidity profiles) that increase the radiative flux differences between the satellite footprint and the POI. We acknowledge these factors, and therefore take them into account in the uncertainty bars of Fig. 9-10.

As mentioned in RC1.1, we would therefore like to stress that the smart sampling approach should not be used deterministically, meaning that it should indeed not be used as a tool to infer cloud conditions at the POI based on an instantaneous satellite footprint farther away. We rather developed a statistical approach, which includes statistical variability in the uncertainty estimates, based on all factors that influence the radiative flux differences between locations.

Under a highly variable atmosphere, differences between the POI and the location of sampling will increase faster with distance compared to when atmospheric conditions are more homogenous. However, given the relatively small amount of samples, limiting the distance under such conditions would decrease the representativeness of the derived radiative fluxes for the heterogeneity of the atmosphere, since it would only represent a very brief snapshot near the POI, while conditions would be rapidly changing under such atmosphere. By sampling at larger distances, this heterogeneity will be better represented in the retrieved fluxes, which are then more like an average or representative state for the variable atmosphere.

Please see RC1.1 for details on where we have included this information in the revised manuscript.

*RC2.2 The main results in this work focus on a few POIs to demonstrate its effectiveness. While it is mentioned in the text that a large-scale extension of the sampling approach is beyond the focus of the manuscript, it would be useful to understand how practical it would be to extend this to a larger scale. I am not sure how difficult it would be to test this method on a gridded surface. It would be great if an example could be included to show how a larger sample would compare to the FLXHR-LIDAR product. If this is not possible I feel it is necessary to mention if this methodology computationally efficient to be applied at a larger scale or simple enough for a user to implement on their own.*

**We agree that it would certainly be interesting to apply the methodology on a wider scale, and this should definitely be the next step in this research. However, we did not include this in this study, due to the very different sampling mechanisms that arise when applying the method to obtain averages in a grid box, instead of using the method for estimating radiative fluxes at one particular point in the polar regions, which is the aim of the present study.**

**What is currently often done for gridded products (see for example Van Tricht et al., 2016), is to take all satellite tracks and footprints that intersect with a gridbox, to calculate the average radiative fluxes for that gridbox. While we believe that many of the components of the smart sampling approach (such as the spatial correction procedure) could benefit the gridded product, and the computational requirements would certainly not be a bottleneck issue, the core mechanism of the sampling should be completely revised. With the current method –a circular sampling around a POI, say the center point of the gridbox -, there would inevitably be oversampling of satellite observations with neighbouring gridboxes, violating the indepency requirement of the individual satellite footprints, running the neighbouring gridboxes dependant from one another. This should be thoroughly researched before a reliable gridded smart sampling method can be developed. We suggest to add this information to the discussion, but keep the current smart sampling approach aimed at estimating radiative fluxes for a local POI:**

*However, although there are no computational limitions for the method to be applied on a large-scale grid, the current method would inevitably result in oversampling of satellite observations between neighbouring gridboxes, violating their independence. This should be thoroughly researched before a reliable gridded version of the smart sampling approach can be developed.*

**Specific comments**

*RC2.3 Pg 3, Line 15: I am not sure if Figure 1 is necessary. I think a description of the errors is sufficient.*

**It is true that this concept might be trivial to some readers. However, we do think that readers that have no experience in narrow-swath satellite tracks, can benefit from this figure for understanding the basic concepts of sampling frequency and agreement in function of distance from a POI. We would therefore like to suggest to keep the figure in the manuscript.**

*RC2.4 Pg 3, Line 32: How similar are the broadband measurements in the AWS to the 18 bands used in FLXHR-LIDAR?*

**As an example for the AWS sensors, a typical CM3 pyranometer measures SW radiation between 305 nm and 2800 nm, while 2B-FLXHR-LIDAR considers the range 200 nm – 4000 nm. To quantify the impact of this discrepancy, we performed offline runs with the Santa Barbara DISORT Atmospheric Radiative Transfer (SBDART) model, based on a typical sub-arctic summer atmosphere under a solar zenith angle of 0 to minimize the atmospheric scattering. The figure**

below shows that the impact of the different wavelength ranges is significant for TOA SW radiation, but this difference becomes marginal at the surface. We find a difference of 0.8% incident SW radiation at the surface for the two wavelength ranges, demonstrating that that there is a negligible wavelength impact on the SW retrievals by the AWS and 2B-FLXHR-LIDAR.

With regard to downwelling LW radiation, the CG3 pyrgeometer starts measuring at 5 μm, while 2B-FLXHR-LIDAR starts at 4 μm. However, integrating the Planck function for both wavelength ranges results in a difference of <1%.

We conclude that the slightly different wavelengths for the AWS stations and the 2B-FLXHR-LIDAR algorithm do not significantly impact the retrievals, and therefore our comparisons. We added this information to Sect. 2.2 in the revised manuscript:

*The broadband SW fluxes cover the wavelengths 200-4000 nm, while the LW fluxes cover the range 4-50 μm. These ranges are slightly different from what is measured by the AWS sensors in the field. For example, a typical CM3 pyranometer measures SW radiation between 305 and 2800 nm, and a CG3 pyrgeometer measures LW radiation from 5 μm onwards. We performed offline radiative transfer model runs under a typical Arctic atmosphere, to quantify the impacts of the differences in these ranges between AWS sensors and the 2B-FLXHR-LIDAR algorithm. For both downwelling SW and LW radiative fluxes, differences are below 1%, demonstrating that these wavelength range differences do not significantly impact the retrievals.*

[Figure]

*RC2.5 Pg 4, Line 3: Include the acronym for CPR*

**Corrected.**

*RC2.6 Pg 4, Line 8: CALIPSO/CloudSat is excellent for cloud detection and aerosol.*

**We added this information to the revised text:**

*CloudSat and CALIPSO were launched in 2006 to globally observe clouds **and aerosols** from a near-polar orbit.*

*RC2.7 Pg 4, Line 9: Are you talking about 2B-FLXHR-LIDAR here? Not all CloudSat products implement auxiliary information.*

**This information is indeed related to the 2B-FLXHR-LIDAR. We agree that this was unclear in the original manuscript. We have revised the text to clarify this as follows:**

*The 2B-FLXHR-LIDAR product used in this study uses CALIOP- and CPR-measured backscattered energy by cloud particles, which are then converted into vertical distributions of cloud ice and liquid water contents and effective radii at a vertical resolution of 240 m, filled in by Moderate Resolution Imaging Spectroradiometer (MODIS) radiance information when the retrieval algorithms of the active sensors fail to converge.*

*RC2.8 Pg 4, Line 13: I think it is worth noting the high vertical resolution of the CloudSat and CALIPSO products. The vertical resolution is one feature that makes the CloudSat/CALIPSO products unique in estimating radiative fluxes.*

**Agree, this information was added in the text (cfr. RC2.7)**

*RC2.9 Pg 5, Line 3: Water bodies emit more radiation compared to what?*

**The following information was added to the revised text:**

*For example, water bodies emit more LW radiation **than snow-covered surfaces**, which warms the atmosphere in addition to higher moisture fluxes as well.*

*RC2.10 Pg5, Line 27: This difference in transmission would be only for clear sky. Are these cases taken only for clear sky measurements?*

**These cases include clouds as well. The correction procedure on transmission assumes equal atmospheric conditions, either clear sky or similar cloudiness. It corrects only for altitude differences. We've added in the revised text that this assumption only holds under similar atmospheric conditions between the two locations. We would like to refer to RC1.1 and RC2.1 for further discussion on how the different atmospheric conditions are treated here.**

*RC2.11 Pg 8, Line 29: The text states yearly averages, however, the caption states monthly.*

**Thank you for mentioning this discrepancy. The figure indeed shows yearly mean biases, which has been changed in the caption accordingly.**

*RC2.12 Pg 9, Line 10: It would be interesting to discuss the impact including the diurnal computations have on the results.*

**Without this correction, the retrievals do not represent daily averages of SW radiation, since they originate from fixed overpass times and according sun positions. We have added this information in the revised text as follows:**

*Due to the fixed overpass times of CloudSat and CALIPSO, the SW radiation retrievals are not representative for the full diurnal cycle of SW radiation. If no correction for this discrepancy were*

*applied, the retrievals would only be valid for the local overpass times and according sun positions of the CloudSat and CALIPSO satellites.*

*RC2.13 Pg 10, Line 18: I do not like the use of the term "sampling frequency" here. While the number of samples available does increase, these are not physical measurements being made and the satellite overpass frequency is not changing.*

**We agree that this confusion should be avoided. We propose to change the terminology on this occasion to** *"sample availability"*

*RC2.14 Line 13, Line 31: Again, the number of samples is increased due to spatial processing. The number of overpasses does not change.*

**We rephrased this part to:**

*Implementing the smart sampling approach is shown to increase on average* **the availability of** *unique satellite overpasses from only two each month for limited-distance sampling < 10 km from the POI to 35 each month, with a consequent increase in total amount of* **available** *satellite samples from 33 to 8,412 (LW) and 7,973 (SW)*

**Reference:**

Van Tricht, K., Lhermitte, S., Lenaerts, J. T. M., Gorodetskaya, I. V, L'Ecuyer, T. S., Noël, B., … van Lipzig, N. P. M. (2016). Clouds enhance Greenland ice sheet meltwater runoff. Nature Communications, 7, art.nr. 10266. http://doi.org/10.1038/ncomms10266s